# Sociability Linked to Reproductive Status Affects Intestinal Parasite Infections in the Red-Billed Chough

**DOI:** 10.3390/pathogens14090915

**Published:** 2025-09-11

**Authors:** Guillermo Blanco, Maria M. Garijo-Toledo, José Luis González del Barrio, Óscar Frías, Raymundo Montoya Ayala, Iñigo Palacios-Martínez

**Affiliations:** 1Department of Evolutionary Ecology, Museo Nacional de Ciencias Naturales, CSIC, José Gutiérrez Abascal 2, 28006 Madrid, Spain; jgonzalezdelbarrio@gmail.com (J.L.G.d.B.); milvusmigrans@gmail.com (Ó.F.); rmontoya@unam.mx (R.M.A.); inigo.palacios@mncn.csic.es (I.P.-M.); 2Vector-Borne Diseases Research Group (ZOOVEC), Department of Animal Production and Health, Public Veterinary Health and Food Science and Technology, Faculty of Veterinary Medicine, Universidad Cardenal Herrera-CEU, CEU Universities, 46115 Valencia, Spain; marilena@uchceu.es; 3Laboratorio de SIG y Análisis Espacial, UBIPRO, Facultad de Estudios Superiores Iztacala UNAM, Av. Los Barrios 1, Los Reyes Iztacala, Tlalnepantla 54090, Estado de Mexico, Mexico

**Keywords:** coccidian, communal roosts, endoparasites, helminths, *Pyrrhocorax pyrrhocorax*, territorial pairs

## Abstract

Social behaviour plays a crucial role in the dynamics of parasitic infections in wild bird populations. The red-billed chough (*Pyrrhocorax pyrrhocorax*), a corvid undergoing notable population declines, shows contrasting social structures linked to reproductive status: non-breeding individuals aggregate in communal roosts during winter, whereas breeding pairs often maintain territorial pair-bonds and roost at nesting sites. This study tested whether differences in sociality (communal roosting vs. territorial pairs) affect intestinal parasite infections. Fresh faecal samples were collected during winter in central Spain and analysed using flotation and McMaster techniques to detect and quantify coccidian oocysts and helminth eggs. The results revealed a relatively high positive rate of *Isospora* (36.2%, *n* = 116) and a low positive rate of helminths (9.5%, *n* = 116) among communally roosting non-breeders, while no parasites were detected in samples from territorial pairs. One communal roost in the Southern Plateau showed higher coccidian positive rate, possibly influenced by structural features that facilitate faecal contact. Although sample size for pairs was limited, the absence of parasites in this group suggests reduced infection risk, likely reflecting superior condition and immune defences rather than differences in exposure alone. These findings highlight the value of non-invasive parasite monitoring as an early-warning tool in wildlife health assessments and stress the importance of considering social behaviour and environmental heterogeneity into conservation strategies for threatened species.

## 1. Introduction

Living in groups increases the risk of transmission and individual exposure to infectious agents, entailing a trade-off in which the benefits of social aggregation come at a potential cost to individual fitness [1,2]. Sociability is shaped by the degree of cohesion among individuals, and the frequency and spatial predictability of shared activities such as foraging, breeding, and resting [3,4]. Host vulnerability to parasites and pathogens is further shaped by biogeographic and evolutionary histories, genetic variability, demographic factors, and interspecific interactions, all of which influence disease dynamics in natural populations [5,6,7]. Understanding these multifaceted dynamics is essential for identifying the ecological and epidemiological drivers behind population declines and biodiversity loss [7]. These factors increasingly interact with intensifying anthropogenic environmental degradation, which impairs host condition and ultimately increases susceptibility to infection [8,9,10], thereby hindering our ability to assess large-scale ecological risks and to design effective conservation strategies for threatened species [11,12,13,14].

Intestinal parasites affect host health depending on multiple intrinsic and ecological factors, including habitat, weather conditions, and diet [15]. For example, hosts in degraded habitats may be exposed to higher parasite loads due to increased contamination, adverse weather conditions can stress individuals and alter their immunity, and diet quality can influence the host’s resistance to infection, thereby affecting overall health and condition [16,17,18]. In addition, sociability may not only facilitate parasite transmission but also influence individual body condition and immune competence, both of which are modulated by age, experience, and reproductive status [19,20,21]. Parasitic infections can also alter condition-dependent host behaviours, leading to changes in social roles and perceived risks by the infected individual or its conspecifics [22,23]. During the pre-reproductive stage, many bird species form flocks of young individuals until they attain reproductive condition, establish pair bonds, and secure territories or nesting sites, subsequently adopting a less social lifestyle [24]. Sociability can vary seasonally, often increasing during winter when many species form flocks and share communal roosts [25]. Socially gregarious species that use communal roosts may be more exposed to parasitic infections than more solitary species [26,27]. However, susceptibility to parasites may not only differ between species, but also between populations and even among individuals within the same population, depending on their specific social strategies and behaviours related to foraging and resting [3,28]. Social bonds, through their positive health effects, may facilitate the maintenance of stable group living [29]. For instance, in cooperatively breeding carrion crows (*Corvus corone*), strong social bonds and relatedness reduced coccidian excretion [30], whereas in non-cooperatively breeding individuals of the same species, only group size and social rank mattered [31], showing that social system differences shape parasite excretion patterns. In some species, these strategies are flexible, with reproductive individuals joining communal roosts in winter or maintaining year-round territories while associating in diurnal foraging groups [25]. The impact of these contrasting behavioural patterns on the incidence of intestinal parasites remains poorly understood in free-living birds.

Given their potential to compromise individual health and population viability, parasitic infections may play a critical role in determining the conservation status of threatened species. They should therefore be carefully integrated into conservation planning and management strategies. Among European corvids, the red-billed chough (*Pyrrhocorax pyrrhocorax*) holds an unfavourable conservation status, with extirpated, increasingly fragmented, and declining populations across much of its range [32]. The impact of parasites on individual health, physiological condition, and population demography remains poorly understood in this species. Haematozoans and ectoparasites, such as mites and lice, have been reported at low prevalence and are generally assumed to exert limited effects on adults [33]. Knowledge about gastrointestinal and other internal parasites remains very limited [34,35,36]. The most comprehensive study to date on helminth and other endoparasitic and pathogenic infections in this species was conducted in the declining population on Islay, Scotland, where juveniles were found to carry heavy burdens of previously unrecorded taxa. These infections affected multiple digestive and respiratory organs and were identified as a major factor limiting post-fledging survival. The impact of helminths in this population is further exacerbated by the occurrence of bacterial infections and severe ocular pathologies, including blindness, which preclude the survival of affected nestlings in the wild [37]. Several factors may contribute to this vulnerability, including environmentally driven dietary shifts toward invertebrate intermediate hosts and reduced immune competence linked to low genetic diversity in small or isolated populations [38]. These findings underscore the urgent need to monitor intestinal infections and associated health impairments in red-billed choughs, while integrating ecological, social, and behavioural data to inform more effective conservation strategies. Nevertheless, it remains difficult to disentangle the effects of exposure due to sociability from the influence of age and status on physical condition and immunocompetence in facing parasites.

This study investigates how sociability influences the occurrence and abundance of intestinal parasites, specifically coccidia and helminths, in the red-billed chough (hereafter, chough). Using non-invasive coprological methods, the hypothesis is tested that individuals engaging in communal roosting behaviour differ in parasitism levels compared to territorial pairs that roost at their nesting sites during winter. Since sociability may enhance parasite transmission, it is predicted that choughs using communal roosts exhibit a higher incidence of intestinal parasites relative to paired birds sleeping at their nesting sites. Differences in parasitism are also examined among communal roosts, including those belonging to the same population as the breeding pairs, as well as between roosts located in two geographically and ecologically distinct areas. This approach provides a meaningful framework for interpreting patterns of parasite occurrence and transmission risk under conditions of environmental differentiation. Given that the red-billed chough is a threatened species, the potential conservation implications of the results are discussed.

## 2. Materials and Methods

### 2.1. Study Areas

The study was conducted in two major regions corresponding to the Northern and Southern Castilian Plateaus, which are separated by the Central System mountain range in central Spain (Figure 1). These areas represent contrasting environmental contexts in terms of climate, altitude, topography, and land cover.

The southern Castilian Plateau lies at an average altitude of approximately 500 to 700 m above sea level and is characterised by relatively flat or gently undulating terrain. The climate is Mediterranean continental, with hot, dry summers and cold winters, with low to moderate annual precipitation (typically 300–500 mm per year). The landscape is predominantly agricultural, consisting of extensive rainfed cereal fields, olive groves, and patches of open scrubland. Soils in the area are often composed of gypsum, which support sparse natural vegetation exploited by foraging choughs. In this region, choughs use a wide variety of structures for both breeding and roosting, including natural cliffs, quarries, farmhouses, and livestock enclosures, both abandoned and in use, as well as industrial ruins, wells, and other man-made constructions [39].

The northern Castilian Plateau is situated at a higher elevation, ranging from 850 to over 1000 m above sea level, and presents a more varied topography due to its proximity to the foothills of the Central System Range. The climate is cooler and more humid than in the Southern Plateau, with higher annual rainfall (400–700 mm per year) and frequent winter frosts. The landscape forms a mosaic of rainfed croplands, rocky outcrops, dry grasslands, and remnant patches of Mediterranean holm oak (*Quercus ilex*) forest, along with other montane woodlands such as pinewoods (*Pinus* spp.), and juniper stands (*Juniperus* spp.). In this region, choughs forage in dry-farmed fields, montane grasslands, and low-growing shrublands typical of high limestone plateaus (páramos). For breeding and roosting, they commonly use natural cliffs, historic buildings such as churches and hermitages, and various human-made structures including bridges and abandoned constructions [40].

### 2.2. Fieldwork

During the winter of 2012–2013 (December–January), faecal samples were collected beneath roosting sites used by choughs. The fieldwork distinguished between two types of individuals: breeding pairs that roosted at their nesting sites and non-breeding individuals that gathered in communal roosts in variable numbers. In total, 139 fresh faecal samples were collected and analysed. Of these, 116 were obtained from individuals using five communal roosts, while 23 samples were collected from breeding pairs roosting solitarily at their nesting sites (*n* = 14). The nesting sites sampled were selected for their ease of access and their dispersed locations across the foraging range used by the flocks forming the communal roosts (Figure 1).

Sampling of breeding pairs was carried out exclusively on the Southern Plateau, at nesting sites located in the provinces of Madrid, Toledo, and Cuenca. Communal roost sampling was conducted in both regions: on the Southern Plateau at two roosts in Madrid province (Aranjuez and La Aldehuela), one in Toledo (Huerta de Valdecarábanos), and one in Cuenca (Barajas de Melo). These communal roosts were located in country houses and livestock enclosures, both abandoned and active, as well as in an anti-aircraft trench from the Spanish Civil War. The number of choughs at each roost varied between 28 and 65 individuals during the sampling period, according to simultaneous counts conducted in January (see census methods in Ref. [41]). On the Northern Plateau, a single roost located in the bell tower of Segovia Cathedral, situated within the historic city centre, in the province of Segovia (Figure 1), was sampled for faeces. This roost comprised approximately 335 choughs during the study period.

Faecal sampling was conducted early in the morning, immediately after the choughs had left their roosts. Plastic sheets were placed on the ground beneath perching sites before sunset and collected at dawn. Since these roosts were exclusively used by choughs, faecal samples could be confidently attributed to this species. Choughs typically perched in a dispersed manner within communal roosts, allowing for the collection of spatially separated samples (at least 50 cm apart) to minimise the risk of sampling the same individual more than once. In the case of breeding pairs roosting at their nesting sites, the repeated use of the same perches allowed for the collection of faeces from both members of the pair, identified by the accumulation of droppings in specific spots. In most cases, two samples were collected per site, corresponding to each partner, although in some instances only one sample could be obtained.

Sample freshness was ensured by selecting faeces based on visual indicators of degradation and desiccation. Sterile gloves were used during collection to prevent contamination. Each faecal sample was divided into two aliquots and preserved separately in vials containing 10% formaldehyde and 2.5% potassium dichromate (K_2_Cr_2_O_7_) (LAMBDA diagnóstico, Valencia, Spain), respectively, for subsequent parasitological analyses.

### 2.3. Parasitological Analysis

Faecal samples were processed using standardised coprological methods. Aliquots preserved in formaldehyde were first homogenised and weighed, then subjected to flotation in a 33% zinc sulphate (ZnSO_4_) solution (LAMBDA diagnóstico, Valencia, Spain) to concentrate lighter parasitic forms such as protozoan oocysts and helminth eggs. The suspension was transferred to McMaster counting chambers for quantification, allowing estimation of the number of parasitic forms per gram of faeces, following standard protocols [42,43,44]. This technique is particularly suitable for detecting and quantifying coccidian oocysts and light-shelled helminth eggs.

To complement this approach, sedimentation techniques were applied to the remaining aliquots to detect heavier parasitic forms, such as trematode eggs and dense nematode eggs that may be missed by flotation. Samples were diluted in water, sieved through a 150 μm mesh, and centrifuged at 1500 rpm for 5 min. The sediment was examined under a light microscope at 10× and 40× magnification.

Aliquots preserved in potassium dichromate were used to assess coccidian sporulation. Subsamples were incubated at room temperature (21–23 °C) for 5–7 days in loosely capped tubes to allow aeration, with periodic stirring to ensure oxygen availability. Sporulated oocysts were identified to the genus level based on oocyst morphology and sporocyst arrangement under light microscopy at 10× and 40× magnifications according to taxonomic keys [45,46].

All coprological analyses were conducted within four weeks of collection to minimise sample degradation. Microscope slides were read by a trained observer to ensure consistency in detection and identification. The rate of infection was defined as the proportion of positive samples per total examined, and infection intensity was estimated from counts of propagules per gram of faeces (oocyst counts per gram: OPG; egg counts per gram: HPG, for coccidia and helminths, respectively) obtained through the McMaster method.

We used the term positive rate rather than prevalence of intestinal parasites to refer to the frequency (%) of positive samples among those analysed, because determining the actual presence or absence (i.e., prevalence) of these parasites in the host is challenging. Several limitations influence detection, especially the typically low parasite loads, the intermittent shedding of eggs or oocysts, and the limited sensitivity of flotation-based methods. This coprological analysis was not intended to characterise the complete community of endoparasites infecting choughs, which would require species-level identification of adult parasites through necropsy and molecular characterisation [37,47], methods beyond the scope of this study. Specifically, DNA-based methods were not employed due to limited material and resources, and because our study was conceived as an exploratory assessment to provide an initial overview of parasitism patterns, which can guide future molecular investigations targeting specific parasite taxa. Instead, the aim was to detect differences in positive rate and intensity of internal parasite infections by using parasite propagule counts as proxies of infection status potentially associated with individual social behaviour.

### 2.4. Statistical Analysis

The shortest credible confidence interval (95% CI) for positive rate (proportion of samples positive for parasites) was calculated as the inverse of the relevant cumulative Beta distribution. Parasite occurrence was compared using contingency tables and Fisher’s exact test. Infection intensity (propagules per gram) was described as mean ± SD and analysed using the non-parametric Kruskal–Wallis test. We used the R package (R Core Team, 2023) cooccur to perform a probabilistic co-occurrence analysis of parasitic taxa, which infers positive, negative, or random associations based on presence–absence data, without relying on predefined metrics, distributions, or randomization procedures [48].

## 3. Results

Parasitological analysis revealed marked differences in parasite occurrence between samples from mated pairs roosting solitarily at their nesting sites and those from communally roosting individuals. None of the samples collected from breeding individuals roosting in pairs (*n* = 23) tested positive for any parasite group (positive rate = 0.0%, CI: 0.0–11.7 across all taxa). In contrast, samples from communal roosting birds (*n* = 116) showed variable positive rate of the different parasite taxa detected.

Coccidian oocysts (*Isospora* sp., Apicomplexa, order Eucoccidiorida) were the most frequently detected parasites (Figure 2), with a positive rate of 36.2% (CI: 27.9–45.1) (42/116) (Figure 3). Helminths were considerably less prevalent: *Capillaria* sp. (Nematoda, order Trichocephalida) eggs were found in 1.7% (CI: 0.3–5.4) of samples (2/116), other nematodes identified to order level (Spirurida and Ascaridida) were found in 5.2% (CI: 2.1–10.2) of samples (6/116), while cestodes belonging to the order Cyclophyllidea were found in 2.6% (CI: 0.6–6.7) of samples (3/116). The overall positive rate of helminths was 9.5% (CI: 5.0–15.7) (11/116), while total parasite rate reached 40.5% (CI: 31.9–49.5) (47/116).

The majority (85.1%; 40/47) of positive samples showed a single parasite taxon, whereas the remaining 14.9% (7/47) carried *Isospora* along with one helminth taxon. No cases of simultaneous infection with multiple helminth taxa were detected. *Isospora* showed the highest co-occurrence rate with nematodes other than *Capillaria* (0.6), whereas co-occurrence rates with Cestoda and *Capillaria* were both 0.2. However, co-occurrence analysis revealed no significant positive or negative associations among co-infecting parasites, indicating that their co-occurrence patterns were consistent with a random distribution.

*Isospora* was the most prevalent parasite genus detected across all communal roosting sites (Figure 3). We found a significant association between roost site and infection status (χ^2^ = 14.36, df = 4, *p* = 0.0061), indicating that *Isospora* was not evenly distributed across sites; the Barajas de Melo roost (ref. E in Figure 1) was excluded from the analysis due to the low number of samples collected (positive rate = 25.0%, CI: 2.6–67.0, *n* = 4), which limited statistical power and increased the risk of unreliable estimates. Pairwise post hoc comparisons using Fisher’s exact tests revealed that Aranjuez roost (ref. A in Figure 1) consistently exhibited a significantly higher proportion of infected individuals compared to the other roosts (Figure 3). Specifically, this site differed significantly from Huerta de Valdecarabanos (*p* = 0.0029), La Aldehuela (*p* = 0.0188), and Catedral de Segovia (*p* = 0.0329). No significant differences were detected among the remaining sites. When pooling data on *Isospora* from roosts in the Southern Plateau, we found no significant difference compared to that of Segovia (Northern Plateau) (Fisher’s exact test, *p* = 0.079). A non-parametric Kruskal–Wallis test was conducted to compare *Isospora* oocyst intensity (OPG) among different roosting sites. The Barajas de Melo roosting site (ref. E in Figure 1), which had only a single extreme value (OPG: 7832), was excluded from the analysis. No statistically significant differences were found in OPG between the remaining roosting sites (H = 7.14, df = 3, *p* = 0.067; Figure 3). Pooling roosts by region, we found no significant difference in *Isospora* OPG between the Northern and Southern Plateaus (Mann–Whitney U test, U = 194.5, df = 1, *p* = 0.88).

*Capillaria* eggs were identified at a low positive rate and were limited to a single roost in the Northern Plateau (Segovia), with HPG indicating moderate infection intensities. Other nematodes showed a low positive rate across three roosting sites in the Southern Plateau. Cestode infections were detected at low rate and confined to a single roost in the Southern Plateau, with variable HPG (Table 1).

## 4. Discussion

The results are consistent with the hypothesis that sociability may influence intestinal parasite infections in the red-billed chough. Specifically, individuals roosting communally during winter, primarily young, non-reproductive individuals, and adult floaters, showed a variable presence of intestinal parasites, including coccidia and helminths, whereas no parasites were detected in samples from territorial pairs. Although the sample size for mated pairs roosting in their nesting sites was limited, it appears sufficient to detect meaningful differences according to published guidelines for studies of this nature [49]. Given that oocysts and helminth eggs are transmitted via the faecal–oral route and can persist in the environment [50], the accumulation of infectious stages in shared sleeping sites used repeatedly over years likely contributes to the observed pattern. This is especially relevant in winter, when birds cluster at high densities in caves, buildings, and other enclosed structures [41], often in close contact, conditions that may promote parasite persistence and transmission. These dynamics illustrate the classic trade-off of social living, where the benefits of aggregation come at the cost of elevated parasite transmission [2], as predicted by theory and supported by evidence from other bird species exposed to increased contact rates and environmental contamination in shared spaces [15].

The detection of *Isospora* in more than a third of communally roosting individuals, contrasted with the complete absence of parasitic forms in paired individuals, highlighting a strong social component in parasite exposure. These intracellular intestinal parasites are transmitted via the faecal–oral route, typically through the ingestion of sporulated oocysts in contaminated environments [15,50,51]. To the best of our knowledge, this is the first report of the presence of this coccidian genus in the chough, although it has previously been recorded in other species within the Corvidae family [52,53,54]. The results revealed clear spatial patterns in the distribution of *Isospora* infections among communal roosting sites, with one location in the Southern Plateau standing out for its consistently higher positive rate. This suggests that local environmental conditions or host dynamics at that site may favour parasite transmission [50,55]. In particular, the structural characteristics of this roosting site, specifically the features of the perching areas used by choughs to spend the night, could influence transmission probability. At this site, choughs roost on vertical walls of gypsum cliffs that have been excavated by humans to form a narrow trench. This configuration results in choughs perching at different vertical levels, thus increasing the likelihood of defecation directly onto lower perches or other individuals. This behaviour has been frequently observed during roost monitoring and confirmed during chough capture events for banding at this site [56], where individuals often exhibit faecal staining on their plumage, potentially facilitating the ingestion of parasite propagules during preening. This type of roosting structure is not unique to this location but also occurs in other roosting sites such as wells, chasms, crevices, and cliff faces. Related to this, it has been documented that younger or subordinate birds tend to occupy the more exposed, lower perches, while dominant adults roost higher up [57]. Such a vertical arrangement may increase the risk of infection through greater contact with faeces from conspecifics, potentially explaining the higher coccidia rate at this site compared to others, where roosting usually occurs on single-level horizontal beams inside abandoned houses or barns (see examples in Ref. [41]). This pattern also suggests that subordinate or younger individuals could be exposed to a higher risk of parasitism due to increased contact with conspecific faeces. Therefore, the structure of roosting sites and their influence on exposure to faecal matter may represent an overlooked factor affecting infection risk in birds.

Despite differences in *Isospora* positive rate across sites, no significant variation was found in infection intensity, either among roosts or between broader geographic regions, suggesting that hosts may be able to regulate parasite proliferation regardless of transmission risk. The positive rate found in our study is high, though not exceptional in contexts of high sociability and density, as supported by published data on wild birds [18,47,50,54,58,59]. In avian populations, coccidian loads can vary widely, ranging from tens to several thousand oocysts per gram of faeces in severe infections [50,51,60,61]. The high variability observed in our dataset suggests that some individuals carry high parasite loads (peaks), while others present low or negligible levels. Generally, counts below 100 OPG are considered low to moderate and often subclinical, whereas counts in the hundreds or thousands tend to be associated with clinical symptoms such as diarrhoea, nutrient malabsorption, and general debilitation [15,50,51,60,61,62]. Therefore, the infection intensities observed in our study indicate the presence of parasites but not massive or acute infections.

Other parasite taxa, such as *Capillaria*, other nematodes, and cestodes, were detected only sporadically and at low rate, often limited to single roosts or regions. The total positive rate of helminths detected in our study falls within the low-to-moderate range reported for other wild bird species and aligns with generally healthy populations without mass infestations. Helminths can cause intestinal damage, weight loss, or anaemia in case so heavy infections [15,63]. The low values found in our study suggest instead to light or early-stage infections. For instance, *Capillaria* and other nematode infections around 50–100 HPG are typically considered moderate to high and potentially pathogenic [64]. Likewise, cestode intensity was low, and while cestodiasis in birds can show highly variable burdens, the values detected here are indicative of mild or poorly established infections [15,65]. This points to endemic, sustained infections and a likely equilibrium between host and helminths, with little evidence of severe clinical disease in the studied populations.

The fact that samples were analysed from both members of mated pairs roosting at their nesting sites adds robustness to the results, as it allows for replication of the environmental conditions experienced by the two bonded individuals. It is noteworthy that reproductive pairs maintain lifelong bonds, frequently roost in physical contact at their nesting sites [66], engage in mutual preening, and that males feed females year-round, particularly prior to laying and during incubation. Age and reproductive status are also key mediators of parasitism. Young, non-breeding birds not only display more gregarious behaviour [67,68] but may also possess immature or less efficient immune systems, making them more susceptible to infection and less capable of controlling parasite loads. Furthermore, nutritional stress or lower body condition in juveniles and subadults, compared to adults [69,70], may compromise their resistance and increase the likelihood of infection. These age-related vulnerabilities may help explain the higher rate of infections observed in communally roosting individuals, although the cross-sectional nature of our study limits the ability to directly assess individual condition or immune function.

The absence of infection in territorial pairs may be attributable to several non–mutually exclusive mechanisms: (i) reduced exposure resulting from solitary roosting and lower faecal accumulation at nesting sites; (ii) acquired immunity in older, reproductively active individuals; (iii) higher-quality or distinct diets associated with territory ownership; and/or (iv) health benefits derived from the high-quality, long-lasting bonds between reproductive partners. While the small sample size of the paired group warrants caution, the complete absence of parasites in territorial pairs strongly suggests a real pattern of reduced parasite risk. In the southern plateau, both communal roosts and territorial pairs were sampled, which allowed us to compare both types of sites. In this region, parasite infections were only detected in birds from communal roosts, not in territorial pairs. In contrast, in the northern plateau we only had access to a single communal roost, so no territorial pairs were sampled there. Therefore, the apparent absence of parasites in the territorial pairs does not mean that communal roosts are the only places where infections occur, but rather that in the southern plateau the difference between roosting and nesting sites was evident. Although exposure to direct contact with conspecifics is lower in territorial pairs, this alone cannot fully explain the lack of transmission, especially if parasite acquisition occurs at shared foraging grounds. Territorial pairs often forage in the same flocks as non-reproductive individuals from communal roosts, particularly in winter [67,68]. Moreover, breeding pairs occasionally join communal roosts to spend the night with non-territorial birds [69]. In both scenarios, reproductive adults may be exposed to the same parasites as the infected non-breeders. These observations support the hypothesis that the apparent absence of intestinal parasites in reproductive pairs is more likely attributable to superior physical condition, enhanced immune defences, and the benefits of enduring pair bonds, rather than to differences in exposure or ecological factors alone.

Our findings raise concerns about the potential differences in health impacts of digestive parasites between and pre-breeding and mated choughs. Although many infections may be subclinical, they can impair nutrient absorption, growth, and immune function, ultimately reducing survival and recruitment [15,63]. This is especially critical for small, isolated populations where juvenile survival may be a limiting demographic factor. One limitation of our study is the lack of more precise taxonomic identification of the detected parasites, which would have improved our understanding of their epidemiology and ecological relevance. However, unlike the Scottish population studied on Islay, and considering the taxa identified in our study, we have not observed clinical symptoms of parasitism as severe as those reported there [37]. Notably, after handling several thousand nestlings, we have never observed a blind individual, a condition that on Islay has been linked to reduced genetic variability [71]. Similarly, respiratory symptoms linked to gapeworm (*Syngamus trachea*) infection [34,35,37] have not been documented in our study population, nor in others across mainland Spain and the Canary Island of La Palma, after handling several thousand full-grown individuals for ringing and research [41]. This may reflect the comparatively high genetic diversity of the Iberian and Canarian populations, which likely acted as refugia during Pleistocene glacial bottlenecks [72,73]. In contrast, northern European populations appear to be genetically impoverished remnants of postglacial dispersal from Iberian refugia [72,74]. These findings suggest a potential connection between biogeography, genetic diversity, and parasitism as influential factors in avian demography, dynamics, and conservation status, an aspect that remains underexplored in the management of many threatened bird species.

This study underscores the value of non-invasive parasite monitoring as an early warning tool in wildlife health assessments, highlighting the importance of incorporating social behavioural differences and environmental heterogeneity into conservation and population management frameworks for threatened species. Potential strategies include regular collection of faecal samples at communal roosts and during foraging, combined with molecular and microscopic analyses, as well as seasonal sampling to evaluate changes due to environmental conditions and the life cycles of both parasites and hosts. We note as limitations that sampling exclusively in winter may not capture temporal variation in parasite rate, and that low parasite loads in solitary roosts could remain undetected with traditional methods, suggesting that more sensitive molecular techniques may provide a more accurate assessment.

## 5. Conclusions

This study provides preliminary evidence that social behaviour is a key factor shaping digestive parasite risk in the chough. The results underscore the importance of incorporating sociability associated with age and reproductive status into conservation planning, particularly for species in which early-life health challenges may have out-sized demographic consequences. Future studies should aim to confirm these patterns across broader spatial and temporal scales, ideally through longitudinal monitoring of marked individuals, to disentangle the roles of age, social status, condition, and behaviour in parasite dynamics. Research should further assess the functional effects of infection on individual performance and fitness and explore how environmental and genetic factors interact with sociability to shape host–parasite interactions in different populations of this threatened species.

## Figures and Tables

**Figure 1 pathogens-14-00915-f001:**
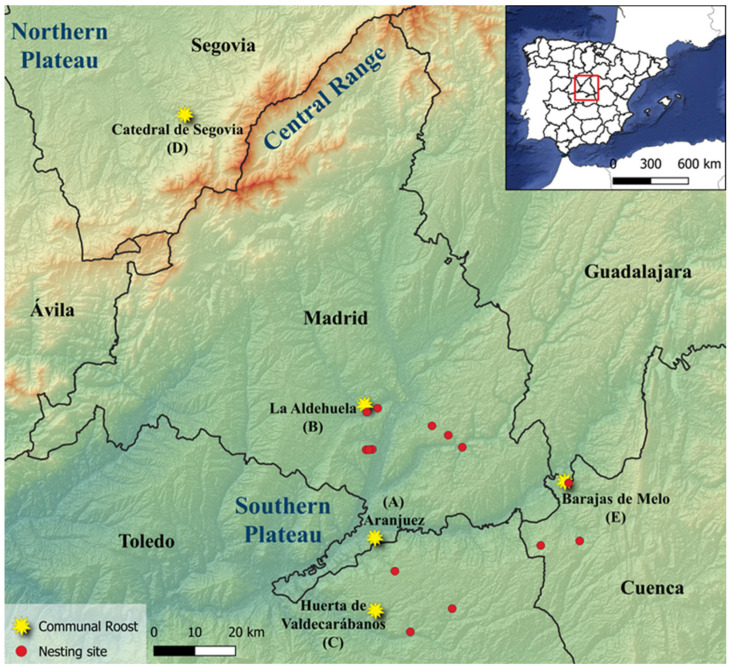
Geographic distribution of communal roosts (yellow stars) and nesting sites (red circles) of red-billed choughs (*Pyrrhocorax pyrrhocorax*) sampled for faecal analyses in central Spain. Locations are grouped into two distinct biogeographic subregions (the Northern and Southern Castillian Plateaus) separated by the Central Range. Sampled sites are displayed on a shaded relief map with provincial boundaries. The inset shows the location of the study area within the Iberian Peninsula.

**Figure 2 pathogens-14-00915-f002:**
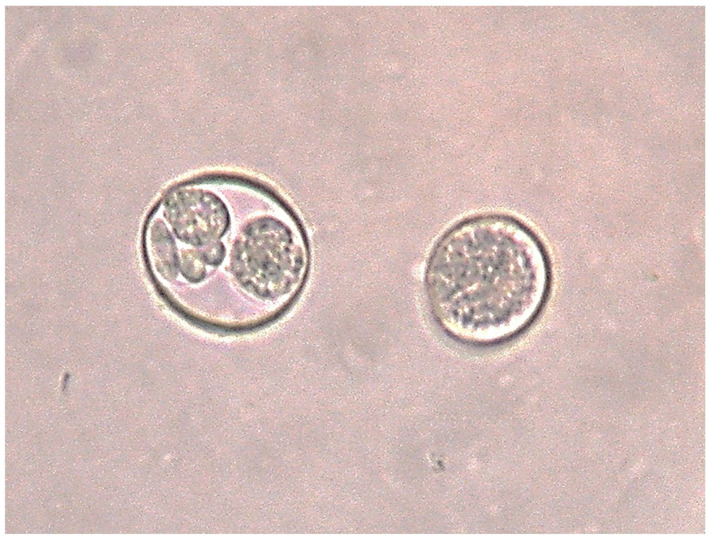
Sporulated (left) and non-sporulated (right) *Isospora* spp. oocysts from a red-billed chough (*Pyrrhocorax pyrrhocorax*) faecal sample preserved in 2.5% potassium dichromate (K_2_Cr_2_O_7_), 40× magnification.

**Figure 3 pathogens-14-00915-f003:**
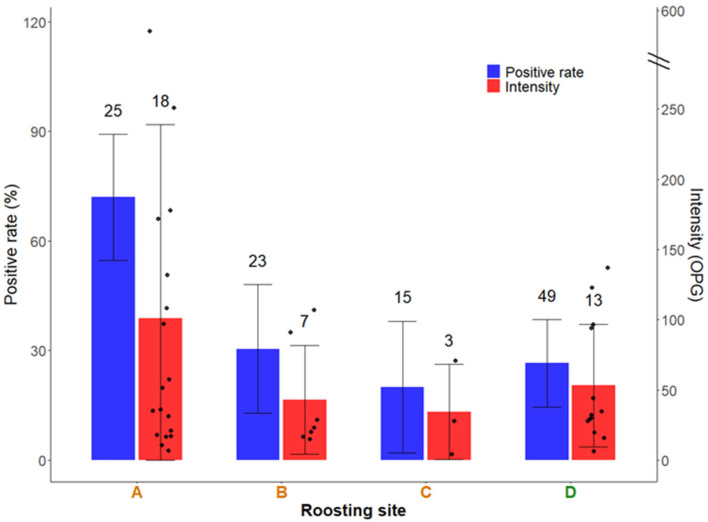
Positive rate and infection intensity of *Isospora* sp. in faecal samples from red-billed choughs (*Pyrrhocorax pyrrhocorax*) collected at communal roosting sites in central Spain. Shown are Isospora rate (%, 95% CI) and mean ± standard deviation (SD) oocyst counts per gram (OPG). The number of faecal samples analysed, and the number of positive samples are shown above the bars. Only infected samples were used to calculate OPG values. Sampling roosting sites in the Southern Plateau (references A, B, and C, letters in orange colour) and Northern Plateau (reference D, letter in green colour) refer to distinct biogeographic subregions within the study area (see Figure 1).

**Table 1 pathogens-14-00915-t001:** Positive rate and infection intensity of helminths. in faecal samples from red-billed choughs (*Pyrrhocorax pyrrhocorax*) collected at communal roosting sites in central Spain. Shown are the number of faecal samples analysed (n), prevalence (%, 95% confidence interval: CI), and mean egg counts per gram (HPG ± SD), with the observed range in parentheses. Only infected samples were used to calculate HPG values. The Southern and Northern Plateaus refer to distinct biogeographic subregions within the study area (see Figure 1).

		*Capillaria* sp.	Other Nematodes	Cestodes
Roosting Site	*n*	Positive Rate (95% CI) HPG (Range)	Positive Rate (95% CI) HPG (Range)	Positive Rate (95% CI) HPG (Range)
*Southern Plateau*				
Aranjuez	25	0.0 (0.0–10.9)	16.0 (5.4–33.0) 48.0 ± 50.5 (7–134), *n* = 4	0.0 (0.0–10.9)
La Aldehuela	23	0.0 (0.0–11.7)	4.3 (0.2–18.4) 5.0, *n* = 1	0.0 (0.0–11.7)
Huerta de Valdecarábanos	15	0.0 (0.0–17.1)	6.7 (0.4–26.6) 6.0, *n* = 1	20.0 (5.7–43.1) 58.0 ± 18.8 (28–74), *n* = 3
Barajas de Melo	4	0.0 (0.0–45.1)	0.0 (0.0–45.1)	0.0 (0.0–45.1)
*Northern Plateau*				
Catedral de Segovia	49	4.1 (0.7–12.3) 40.0 ± 28.3 (20–60), *n* = 2	0.0 (0.0–5.8)	0.0 (0.0–5.8)

## Data Availability

The original data presented in the study are openly available in Zenodo. https://doi.org/10.5281/zenodo.16812733.

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
