# Peer review of "Sociability Linked to Reproductive Status Affects Intestinal Parasite Infections in the Red-Billed Chough"

_pathogens, 2025, doi:10.3390/pathogens14090915_

Round 1

Reviewer 1 Report

Comments and Suggestions for Authors

The authors have conducted a very interesting study on how sociability, linked to reproductive status, influences intestinal parasite infections in the red-billed chough. The manuscript is generally well written and addresses an important gap in the literature regarding this potentially endangered species. However, several aspects require revision, namely:

  1. Overall, the repeated use of expressions such as “we tested…,” “we investigated…,” and “we examined…” gives the manuscript a somewhat informal tone. The text would read more professionally if these were replaced with neutral constructions. For example, instead of writing “We tested the hypothesis that differences in sociality (communal roosting vs. territorial pairs) affect intestinal parasite infections,” it would be preferable to write “This study tested the hypothesis that differences in sociality (communal roosting vs. territorial pairs) affect intestinal parasite infections.” Similar revisions could be applied throughout the manuscript.
  2. Abstract

2.1. The abstract could be made a little shorter. A more focused summary that highlights the main findings would make it even clearer.

2.2. In lines 24–26 (“Results showed a relatively high prevalence of Isospora and a low prevalence of helminths…”), the description of the results could be strengthened by including the actual values or percentages, rather than only qualitative terms.

  1. Introduction

3.1. Line 61–63. “Intestinal parasites affect host health depending…” Although the authors clearly specify the factors that influence host health, they do not explain how these factors actually influence. Providing brief examples or mechanisms (e.g. how diet or weather conditions impact parasite loads and, consequently, host condition) would strengthen this section.

  1. Material and Methods

4.1. line 167 and 168– Authors describe that 116 samples were obtained from five communal roosts but doesn’t clarify how many samples were from each root. Same with the 23 samples from solitary rooting. This information could be done in table form for better understanding.

4.2. It appears that one nesting site is missing from Figure 1, as only 13 out of the 14 sites are visible on the map. The authors may want to check the figure and ensure that all sites are accurately represented.

  1. Discussion

5.1. Lines 315–319. Intestinal parasite infection is usually higher in young individuals due to their more fragile immune systems and because the infection route is faecal–oral, a potential link to sociability is plausible. However, it might be helpful for the authors to clarify how much of the higher prevalence in communal roosts can be attributed to sociability versus the age structure of the individuals. As noted in the discussion (lines 414 and 428–430), the absence of parasites in solitary roosting birds may reflect “acquired immunity in older, reproductively active individuals” or “superior physical condition, enhanced immune defenses, and the benefits of enduring pair bonds, rather than differences in exposure alone”. Communal roosts, in contrast, were “primarily young, non-reproductive individuals, and adult floaters,” who would be expected to have higher prevalence from the outset. A brief acknowledgment of this factor would strengthen the interpretation of the results.

5.2. line 315 Should be toned down

5.3. Author acknowledges the importance of “value of non-invasive parasite monitoring as an early warning tool in wildlife health assessments” line 431-432. It would be useful to expand on this by briefly outlining potential strategies for implementing such monitoring and how they could be applied specifically to this threatened species. This could provide practical guidance for conservation and management efforts.

5.4. line 339 – reference “Liu et al., 2029” the year in wrong

5.5. The authors could mention as a limitation that seasonal monitoring, rather than sampling exclusively in winter, would provide a more complete understanding of parasite prevalence across different times of the year. Another limitation relates to the absence of gastrointestinal parasites in solitary roosts. It is possible that low parasite loads could have influenced detection, as traditional parasitological methods may fail to identify infections with very low intensity. The authors might consider noting that more sensitive molecular methods could provide a more accurate assessment of parasite presence.

  1. Results

6.1. line 278 – In the figure legend, there appears to be an extra period at the end. Removing it would correct the punctuation.

6.2. Does the size of the communal roost influence the prevalence of gastrointestinal parasites? If you have this information would be interesting

Author Response

Reviewer 1

The authors have conducted a very interesting study on how sociability, linked to reproductive status, influences intestinal parasite infections in the red-billed chough. The manuscript is generally well written and addresses an important gap in the literature regarding this potentially endangered species.

Authors’ response: We sincerely thank the reviewer for the positive and encouraging comments.

However, several aspects require revision, namely:

Overall, the repeated use of expressions such as “we tested…,” “we investigated…,” and “we examined…” gives the manuscript a somewhat informal tone. The text would read more professionally if these were replaced with neutral constructions. For example, instead of writing “We tested the hypothesis that differences in sociality (communal roosting vs. territorial pairs) affect intestinal parasite infections,” it would be preferable to write “This study tested the hypothesis that differences in sociality (communal roosting vs. territorial pairs) affect intestinal parasite infections.” Similar revisions could be applied throughout the manuscript.

Authors’ response: We have revised the manuscript accordingly, replacing expressions such as “we tested…,” “we investigated…,” and “we examined…” with neutral constructions as suggested.

Abstract

2.1. The abstract could be made a little shorter. A more focused summary that highlights the main findings would make it even clearer.

Authors’ response: We appreciate this suggestion and have shortened the abstract to provide a more focused summary of the main findings.

2.2. In lines 24–26 (“Results showed a relatively high prevalence of Isospora and a low prevalence of helminths…”), the description of the results could be strengthened by including the actual values or percentages, rather than only qualitative terms.

Authors’ response: We have included the specific prevalence values as suggested by the reviewer.

Introduction

3.1. Line 61–63. “Intestinal parasites affect host health depending…” Although the authors clearly specify the factors that influence host health, they do not explain how these factors actually influence. Providing brief examples or mechanisms (e.g. how diet or weather conditions impact parasite loads and, consequently, host condition) would strengthen this section.

Authors’ response: We thank the reviewer for this helpful suggestion. We have revised the manuscript to provide brief examples of how the listed factors influence host health via parasite loads. Specifically, we now explain how habitat quality, weather conditions, and diet can affect parasite abundance and, consequently, host condition.

Material and Methods

4.1. line 167 and 168– Authors describe that 116 samples were obtained from five communal roosts but doesn’t clarify how many samples were from each root. Same with the 23 samples from solitary rooting. This information could be done in table form for better understanding.

Authors’ response: We thank the reviewer for this observation. The sample size for each communal roost is already provided in Figure 2 and Table 1, while the sample size for territorial pairs is specified in the text, corresponding to 1–2 samples per nest.

4.2. It appears that one nesting site is missing from Figure 1, as only 13 out of the 14 sites are visible on the map. The authors may want to check the figure and ensure that all sites are accurately represented.

Authors’ response: We thank the reviewer for pointing this out. One nesting site appears partially overlapped with another in Figure 1, but all 14 sites, including the territorial pairs, are represented. Zooming in on the figure makes it clear that 14 points are present.

Discussion

5.1. Lines 315–319. Intestinal parasite infection is usually higher in young individuals due to their more fragile immune systems and because the infection route is faecal–oral, a potential link to sociability is plausible. However, it might be helpful for the authors to clarify how much of the higher prevalence in communal roosts can be attributed to sociability versus the age structure of the individuals. As noted in the discussion (lines 414 and 428–430), the absence of parasites in solitary roosting birds may reflect “acquired immunity in older, reproductively active individuals” or “superior physical condition, enhanced immune defenses, and the benefits of enduring pair bonds, rather than differences in exposure alone”. Communal roosts, in contrast, were “primarily young, non-reproductive individuals, and adult floaters,” who would be expected to have higher prevalence from the outset. A brief acknowledgment of this factor would strengthen the interpretation of the results.

Authors’ response: We thank the reviewer for this insightful comment and agree that distinguishing between the effects of sociability and intrinsic factors such as age and reproductive status is challenging. As suggested, we have clarified this point in the Discussion, by including a sentence explicitly acknowledging the difficulty of disentangling the effects of sociability from the influence of age and status on physical condition and immunocompetence in facing parasites.

5.2. line 315 Should be toned down

Authors’ response: We thank the reviewer for this comment. The statement has been toned down accordingly.

5.3. Author acknowledges the importance of “value of non-invasive parasite monitoring as an early warning tool in wildlife health assessments” line 431-432. It would be useful to expand on this by briefly outlining potential strategies for implementing such monitoring and how they could be applied specifically to this threatened species. This could provide practical guidance for conservation and management efforts.

Authors’ response: We thank the reviewer for this valuable suggestion. We have expanded the manuscript to briefly outline potential strategies for implementing non-invasive parasite monitoring, including regular faecal sampling at communal roosts and during foraging, coupled with molecular and microscopic analyses, as well as seasonal sampling to evaluate changes due to environmental conditions and the life cycles of both parasites and hosts. We have also moved this section into a separate paragraph to improve the continuity and flow of the text. We indicate how these strategies could be applied specifically to this threatened species to inform conservation and management decisions.

5.4. line 339 – reference “Liu et al., 2029” the year in wrong

Authors’ response: We thank the reviewer for pointing this out. This was a typographical error, and the correct year for the reference “Liu et al.” is 2019.

5.5. The authors could mention as a limitation that seasonal monitoring, rather than sampling exclusively in winter, would provide a more complete understanding of parasite prevalence across different times of the year. Another limitation relates to the absence of gastrointestinal parasites in solitary roosts. It is possible that low parasite loads could have influenced detection, as traditional parasitological methods may fail to identify infections with very low intensity. The authors might consider noting that more sensitive molecular methods could provide a more accurate assessment of parasite presence.

Authors’ response: We thank the reviewer for these additional valuable comments. We note that the importance of seasonal sampling to evaluate changes in parasite prevalence was already addressed in a previous response. In the revised manuscript, we have now explicitly included this limitation, noting that sampling exclusively in winter may not capture temporal variation in parasite prevalence. Additionally, we have acknowledged that the absence of detected gastrointestinal parasites in solitary roosts could reflect low parasite loads that are difficult to detect with traditional parasitological methods, and that more sensitive molecular techniques could provide a more accurate assessment of parasite presence.

Results

6.1. line 278 – In the figure legend, there appears to be an extra period at the end. Removing it would correct the punctuation.

Authors’ response: Done.

6.2. Does the size of the communal roost influence the prevalence of gastrointestinal parasites? If you have this information would be interesting

Authors’ response: We thank the reviewer for this interesting question. We note that our study only included sampling in five communal roosts, which limits the statistical power to analyze the effect of roost size on parasite prevalence. Nevertheless, a superficial evaluation of the available data does not suggest any clear relationship between the number of individuals in a roost and gastrointestinal parasite prevalence

Reviewer 2 Report

Comments and Suggestions for Authors

In the reviewed MS a group of scientists from Spain and Mexico compared prevalence of intestinal parasites in two groups of the red-billed chough, differing in their social behavior: (1) non-breeding individuals aggregate in communal roosts during winter and (2) breeding pairs which maintain territorial pair-bonds and roost at nesting sites. The authors collected bird feces in two geographically separated mountainous areas in Spain and detected presence of parasites (mainly worms and coccids) in lab conditions. The results confirmed the primary hypothesis of the study that in the red-billed chough sociability linked to reproductive status affects intestinal parasite infections. The MS is well-written. All references are relevant. The statistical methods were used and results were interpreted correctly. I have no major criticism on this MS and consider it close o acceptance in its current form providing the authors address several minor remarks below.

References: please, follow the journal forma for all references in the MS

80-81 For instance, differences in the social systems of carrion crows (Corvus corone) may lead to varying associations with coccidian excretion patterns (Wascher, Canestrari, & Baglione, 2019; Wascher, 2021). – could you briefly describe how exactly it “leads to varying associations with coccidian excretion patterns”? This would be interesting to compare with what you observed for Isospora and coccidians

278 (see Fig. 1).. – remove the extra dot

Table 1. Please confirm that all CI for prevalence are correct. E.g.: 16.0 (0.05-0.33), should it be 5-33?

Could you please insert a brief text in the M&M section explaining why you did not use DNA-based methods?

Author Response

Reviewer 2.

In the reviewed MS a group of scientists from Spain and Mexico compared prevalence of intestinal parasites in two groups of the red-billed chough, differing in their social behavior: (1) non-breeding individuals aggregate in communal roosts during winter and (2) breeding pairs which maintain territorial pair-bonds and roost at nesting sites. The authors collected bird feces in two geographically separated mountainous areas in Spain and detected presence of parasites (mainly worms and coccids) in lab conditions. The results confirmed the primary hypothesis of the study that in the red-billed chough sociability linked to reproductive status affects intestinal parasite infections. The MS is well-written. All references are relevant. The statistical methods were used and results were interpreted correctly. I have no major criticism on this MS and consider it close o acceptance in its current form providing the authors address several minor remarks below.

Authors’ response: We sincerely thank the reviewer for the positive and encouraging comments on our manuscript. We greatly appreciate the time and effort devoted to evaluating our work.

References: please, follow the journal forma for all references in the MS

Authors’ response: We have revised all references in the manuscript to comply with the journal’s formatting requirements in the updated version.

80-81 For instance, differences in the social systems of carrion crows (Corvus corone) may lead to varying associations with coccidian excretion patterns (Wascher, Canestrari, & Baglione, 2019; Wascher, 2021). – could you briefly describe how exactly it “leads to varying associations with coccidian excretion patterns”? This would be interesting to compare with what you observed for Isospora and coccidians

Authors’ response: We thank the reviewer for this suggestion. We have expanded the text to better explain how differences in social organization influence coccidian excretion in carrion crows (Corvus corone), highlighting that in cooperatively breeding groups, strong social bonds and relatedness reduce excretion, while in non-cooperatively breeding groups, group size and social rank are the main factors. This clarification allows a clearer comparison with our observations for Isospora and other coccidians.

278 (see Fig. 1).. – remove the extra dot

Authors’ response: Done

Table 1. Please confirm that all CI for prevalence are correct. E.g.: 16.0 (0.05-0.33), should it be 5-33?

Authors’ response: We thank the reviewer for pointing this out. Indeed, the reviewer is correct: prevalence is reported in percentage, while the confidence intervals were previously expressed as proportions. We have corrected this error both in the text and in Table 1, so that the CIs now correspond correctly to percentages.

Could you please insert a brief text in the M&M section explaining why you did not use DNA-based methods?

Authors’ response: We thank the reviewer for this suggestion. We did not use DNA-based methods due to limited material and logistical resources. Moreover, our study is intended as an exploratory, preliminary assessment to provide an initial overview of parasitism patterns, which can later be followed by more detailed molecular studies focused on specific taxa. We have included a sentence addressing this point in the Materials and Methods section

Reviewer 3 Report

Comments and Suggestions for Authors

The MS is an interesting story related with socical activity associated with reproductive status un parasite inection. The following aspects can be applied to improve the quarlity of the MS.

Abstract: 
1. accurate data for parasite infection and species for parasites, and statistical analysis should be used in this part to compare parasite infection in those two groups. 
2. except for Isospora, any other coccidian species were found. If yes, indicate them.

Results:
1. how to differential Capillaria from other helminths in Line 256.
2. as indicated in Figure 1, some communal roost and nesting site are located in the same regions, so why parasite were found only in communal roost, but not in communal roost.
3. percentages for infection should be compared to know if statistical significance exist by p value.
4. what's the meaning of number upon each column.
5. morphological evidence should be supplied, thus a figure to show parasite found in this study is needed.
6. prevalence should be replaced by positive rate in the whole MS, since only limited samples were tested.
7. there should be a table to summary parasites, positive rate and intensity found in this study.

Author Response

Reviewer 3

The MS is an interesting story related with socical activity associated with reproductive status un parasite inection. The following aspects can be applied to improve the quarlity of the MS.

Authors’ response: We thank the reviewer for the positive comments and constructive suggestions to improve the quality of the manuscript.

Abstract:

  1. accurate data for parasite infection and species for parasites, and statistical analysis should be used in this part to compare parasite infection in those two groups.

Authors’ response: We thank the reviewer for this helpful suggestion. As noted in our previous response to Reviewer 1, we have included overall prevalence values in the Abstract, providing quantitative information to strengthen the comparison of parasite infection between the two groups

  1. except for Isospora, any other coccidian species were found. If yes, indicate them.

Authors’ response: We thank the reviewer for this comment. In our study, only Isospora was detected; no other coccidian species were found.

Results:

  1. how to differential Capillaria from other helminths in Line 256.

Authors’ response: We thank the reviewer for this comment. We are not entirely certain whether the question refers to how Capillaria was differentiated from other nematodes. If this is the case, the information is provided in the Materials and Methods section, where we describe the morphological criteria used for identification.

  1. as indicated in Figure 1, some communal roost and nesting site are located in the same regions, so why parasite were found only in communal roost, but not in communal roost.

Authors’ response: We thank the reviewer for this observation. We are not entirely certain about the intended question, as in the southern plateau both communal roosts and territorial pairs were sampled, whereas in the northern plateau only one communal roost was sampled. In the southern plateau, the differences between communal roosts and territorial pairs are clear, which may reflect the combined effects of sociability, condition associated with age and status, or both factors.

  1. percentages for infection should be compared to know if statistical significance exist by p value.

Authors’ response: We thank the reviewer for this comment. In the Results section, statistical comparisons of prevalence and OPG for Isospora between communal roosts are provided, including p-values to indicate significance. However, such comparisons were not possible for helminths due to their occurrence in only a single roost for some taxa, and the generally low prevalence observed in others

  1. what's the meaning of number upon each column.

Authors’ response: We thank the reviewer for this question. We understand that the comment refers to the numbers displayed above each column in Figure 2. If this is the case, these numbers indicate the sample size for each group, as specified in the figure legend.

  1. morphological evidence should be supplied, thus a figure to show parasite found in this study is needed.

Authors’ response: We acknowledge the reviewer’s suggestion. Unfortunately, we do not have high-quality photographs for most parasite taxa detected in this study. However, we included a figure showing Isospora oocysts, as this was the most abundant parasite found. If the editor considers that this image is not sufficiently informative or of adequate quality, we would be open to removing it.

  1. prevalence should be replaced by positive rate in the whole MS, since only limited samples were tested.

Authors’ response: We appreciate the reviewer’s comment. Although the terms occurrence or positive rate could also be applied, we have preferred to keep the term prevalence, since in parasitology it is commonly used to indicate the proportion of positive samples, regardless of parasite abundance. In our study, a sample was considered positive when any parasite was detected, independently of its intensity. Although the overall sample size is relatively limited, it is sufficient to provide a realistic estimate of the proportion of positive samples, particularly for communal roosts where the sample size was larger (n = 116). Moreover, we found no positive samples among territorial pairs, reinforcing the robustness of this pattern. Nevertheless, if the editor considers it necessary, we would have no objection to replacing prevalence with occurrence or positive rate throughout the manuscript.

  1. there should be a table to summary parasites, positive rate and intensity found in this study.

Authors’ response: We thank the reviewer for this suggestion. The requested information is already included in the manuscript: Table 1 summarizes prevalence (positive rate) and intensity data for helminths, while the information for Isospora, the most abundant parasite, is provided in detail in the text and illustrated in Figure 3.

Round 2

Reviewer 1 Report

Comments and Suggestions for Authors

The authors have adressed all the comments suggested, very interesting study.
Hope you the best and tons of success!

Author Response

Reviewer 1

The authors have adressed all the comments suggested, very interesting study. Hope you the best and tons of success!

Authors´response: We sincerely thank you for your kind words and appreciation. We are very grateful for your feedback and best wishes

Reviewer 3 Report

Comments and Suggestions for Authors

The authors have addressed most issues in the previous version, and I have no major concerns, but the following two additional issues need to be revised.

  1. As to "as indicated in Figure 1, some communal roost and nesting site are located in the same regions, so why parasite were found only in communal roost, but not in communal roost", which should be discussed in the MS.
  2. Prevalence should be replaced with positive rate, since limited samples were tested and the limitation of sensitivity for the detection methods.

Author Response

Reviewer 2

The authors have addressed most issues in the previous version, and I have no major concerns, but the following two additional issues need to be revised.

As to "as indicated in Figure 1, some communal roost and nesting site are located in the same regions, so why parasite were found only in communal roost, but not in communal roost", which should be discussed in the MS.

Authors´response: We thank the reviewer for this important comment. To clarify, in our study area the situation differs between regions: in the southern plateau, both communal roosts and territorial pairs were sampled, which allowed us to compare both types of sites. In this region, parasite infections were only detected in birds from communal roosts, not in territorial pairs. In contrast, in the northern plateau we only had access to a single communal roost, so no territorial pairs were sampled there. Therefore, the apparent absence of parasites outside communal roosts does not mean that communal roosts are the only places where infections occur, but rather that in the southern plateau the difference between roosting and nesting sites was evident. We have revised the text of the manuscript to clarify this point and have also included the implications of this pattern in the Discussion section.

Prevalence should be replaced with positive rate, since limited samples were tested and the limitation of sensitivity for the detection methods.

Authors´response: We thank the reviewer for this valuable suggestion. Although in parasitology the term prevalence is commonly used to indicate the proportion of positive samples (irrespective of parasite abundance), we agree that, given the limited sample size and the sensitivity constraints of detection methods, positive rate is more appropriate in this context. Therefore, we have replaced prevalence with positive rate throughout the manuscript, including Figure 2, and have added a clarifying sentence in the Materials and Methods section to explain this choice.